# Cultural Heritage and Religious Phenomenon between Urbicide and Cancel Culture: The Other Side of the Russian–Ukrainian Conflict

Federica Botti * and Cristina Bianchi

Department of Cultural Heritage, Alma Mater Studiorum University of Bologna, 48121 Ravenna, Italy;
cristina.bianchi11@studio.unibo.it
* Correspondence: federica.botti2@unibo.it

**Abstract:** The Russian–Ukrainian conflict, in addition to causing an unacceptable loss of human life, is straining the integrity of Ukraine's cultural heritage, despite the fact that both countries involved are parties to the *1954 Hague Convention for the Protection of Cultural Property in the Event of Armed Conflict* and its *First Protocol*. Churches are one of Ukraine's most important historical assets, as well as symbolic places of Orthodox religious identity common to both the invaders and the invaded. The destruction of these places and their deliberate damage on the part of both sides appear to be part of a more general conflict concerning internal disagreements between Russian and Ukrainian Orthodoxy, which, in turn, reflect two different historical views of the Russian–Ukrainian relationship. A brief reconstruction of relations between the Orthodox Churches operating on the territory of Ukraine demonstrates how religious affiliation has affected the conflict, causing it to become decisive and deeply divisive, so much so that the Patriarchate of Moscow has become an active part of the conflict. This circumstance favours the hypothesis that it is precisely the religious cultural heritage that is most at risk of deliberate destruction. The Russians, by destroying the symbolic places of Ukrainian religious identity (*urbicide*), affirm the spiritual unity of the Russian and Ukrainian peoples. For their part, the Ukrainians attempt to erase the Russian presence and the common religious cultural roots by destroying buildings of worship dear to the tradition of the Moscow Patriarchate (*cancel culture*). They reject the imperial traditions of Russia and, at the same time, claim an independent Church. The question arises as to whether the reconstruction process following the war will take into account the original cultural–religious identities, or whether it will take the opportunity to adopt a new (also) religious identity instead, and whether the old and new instruments offered by law are adequate.

**Keywords:** Orthodox Church; cultural heritage; urbicide; cancel culture; religion; Russian–Ukrainian conflict; nonfungible tokens (NFTs); Patriarchate of Constantinople; intangible heritage; Patriarchate of Moscow; cultural genocide; UNESCO



## 1. Introduction

Although the causes of Russia's aggression against Ukraine are mainly geopolitical, strategic, and economic, linguistic and cultural motivations must also be included, such as the hypothetical identity between Russians and Ukrainians and the religious issues that divide the world of Orthodoxy between the two countries. And it is precisely to these latter, apparently secondary, causes that this paper devotes its attention.

The conflict, which is still ongoing, seems to have the prerequisite of integrating so-called urbicide, meaning the systematic destruction of cities aimed at the annihilation of the culture and identity of the community that undergoing the invasion. More specifically, the attacks on the artifacts of Ukrainian cultural heritage stored in its cities are a product of this destructive intent, despite the existence of well-established international laws protecting cultural heritage that have been ratified by both states involved in the conflict. An examination of this legislation highlights how the deliberate destruction of cultural and

religious property that is not used for military purposes constitutes a war crime under customary international law, both in international armed conflicts and in internal armed conflicts, to the point of being called "cultural genocide".

The attacks against historical monuments, and especially against buildings dedicated to religion, also pose serious problems in terms of the reconstruction of the country's historical–cultural–religious identity, since, once the conflict is over, the reconstruction process could irreparably compromise the multicultural structure that has always characterised Ukrainian society. At present, the religious landscape is characterised by the presence of several minorities, including Muslim Crimean Tatars and Jews who remained after the diaspora. The Catholic community united in the Ukrainian Greek Catholic Church (UGCC) is the third-largest religious denomination in the country, while the Ukrainian Orthodox Church of the Moscow Patriarchate (UOC-MP) is the majority Church in Ukraine. But it is precisely on the Orthodox front where the most critical issues are found. The opposition between the Moscow Patriarchate, under whose aegis the UOC-MP is placed, and the desire for autonomy and independence on the part of other Ukrainian nationalist Orthodox components mirrors, in a religious–identarian aspect, the political claims that animate the conflict between the two countries. On the one hand, Russia, thanks to the symphonic relationship that binds it to the Moscow Patriarchate, intends to place Ukraine in a state of subordination and dependence for the purpose of reaffirming its cultural homogeneity, not only in political and economic, but also in religious terms, which confirms the fact that "the Russian Orthodox Church (ROC) did not hesitate to throw its support behind the Kremlin's war against a neighbouring Orthodox nation" (Luchenko 2023). On the other hand, Ukrainian nationalists also claim an autonomous national identity in religious terms through the recognition of their own autocephalous Orthodox Church, which reflects their values and ideals.

The reasons for this religious clash are rooted in the long-established linkage between Ukraine and Russia, when the metropolis of Kyiv, canonically dependent until the end of the 17th century on the Patriarch of Constantinople, passed to the Moscow Patriarchate (erected in 1589)[1] to become the cultural and religious cradle of modern Russia. This contingency made Ukraine the canonical territory of influence of the Moscow Patriarchate, so much so that the UOC-MP has always constituted the de facto majority Church of the Ukrainian population, regardless of the Russian-speaking members of the ethnic groups settled in the territory. Other nationalistic Orthodox Churches over the centuries have attempted to establish themselves on Ukrainian territory, but the nonrecognition of their autocephaly and an obstructive attitude on the part of the UOC-MP and the Moscow Patriarchate itself prevented them from taking root in the territory in a structured and effectively competitive manner. However, in 2018, the granting of the tomos of autocephaly to the newly established Orthodox Church of Ukraine (UOC, risen from the ashes of the Ukrainian Orthodox Church of the Kyiv Patriarchate (UOC KP)) by the Patriarchate of Constantinople, which effectively added another autocephalous Orthodox-style Church to the country's religious landscape, exacerbated the religious dimension of the Russian–Ukrainian crisis as well. Among other things, the meddling of the Patriarchate of Constantinople has led it to be directly involved in the conflict, since it has been accused by Moscow and its Patriarchate of conniving with the West to further divide the Orthodox Church and, in particular, to damage the Moscow Church. The stance taken by the Patriarchate of Constantinople risks triggering a domino effect against Moscow, which seems to have already begun with the request for a canonical divestment by a community of Dutch believers that abandoned Moscow to embrace the Patriarchate of Constantinople.[2] Even the UOC-MP itself has distanced itself from the Moscow Patriarchate, so much so that it, in the synod held on 21 May 2022, revealed a desire to distance itself from Moscow, despite the fact that its leader, Metropolitan Onuphry, has always been considered close to Moscow Patriarch Kirill.

The World Council of Churches (WCC) itself, the main body in charge of dialogue between the different Christian Churches of the world, has spoken out against Kirill's

support for the war and recommended that the Ukrainian Autocephalous Orthodox Church be granted full WCC membership.

The conflict thus seems to have pervaded the religious plane as well, and this circumstance, if possible, makes the fate of the cultural heritage of religious interest in the invaded country even more uncertain. There is a risk of losing cultural traditions forever in the name of historical revisionism or cancellation of culture in a religious aspect, which could find its legitimacy precisely in the post-war reconstruction of destroyed cities and places of worship. A solution for the preservation of cultural and religious memory may be found in non-fungible tokens (NFT) and in the digitisation of cultural heritage. However, the still-inadequate legislation, economic speculation, and forms of "anaesthetisation" involving the production and preservation of cultural heritage do not guarantee the indispensable re-construction of an identity between *rupture* and *continuity*, nor innovation, nor the knowledge and preservation of material and memories of the past.

It is, therefore, a question of drawing up an initial balance sheet, the conclusions of which cannot but be affected by the ongoing war conflict and the ongoing evolution of ecumenical relations.

## 2. The Russian–Ukrainian Conflict and Its Repercussions on the Integrity of Cultural Heritage: From Urbicide to Cancel Culture

When Putin announced the opening of hostilities with Ukraine on 24 February 2022, the war that ensued from that moment claimed culture, in its quantitatively indefinable and polymorphic forms, among its most helpless and silent victims, from literature to gastronomy; from works of art to cultural assets of religious interest.[3]

The Ukrainian material cultural heritage was the first and most obvious target of Russian bombing when, on the night of 27–28 February, the Ivankiv Historical and Local History Museum (Kyiv) was destroyed, carrying on the contemporary wartime practice of urbicide, a term coined during the Balkan War to designate strategies aimed at targeting urban places which are considered symbolic.[4] In fact, devastating historical, artistic, religious, and cultural heritage seems to be the main purpose of contemporary wars, since it means irreparably striking the enemy's identity as well as their social and cultural values (Mazzucchelli 2010, p. 32). Without dwelling on the operational methods and objectives of the conflict, numerous Ukrainian urban centres have been devastated in both physical and non-material terms. Some of them were swept away for technical or military reasons, but the nature of the destruction caused to others also highlights the symbolism of the offensive action.[5] Indeed, this is a war that affects symbols (both material and immaterial), and besiegers and the besieged bear these values differently (Barattin 2004, p. 333). It is no coincidence that many of the most significant religious and civil monuments, symbols of the history, culture, religious sensibilities, and daily lives of the Ukrainian community, although not military targets, were profoundly damaged after being subjected to a dense bombing rainfall. There have been more than two hundred religious buildings destroyed alone, most of which were Orthodox churches, although there has been no shortage of damage inflicted on mosques, synagogues, Catholic and Protestant churches, institutes of religious education, and important administrative buildings of religious organisations.[6] Supporting the hypothesis of urbicide, so much so as to bring to mind the controversial term "cultural genocide",[7] is the fact that the first data collected on the war crimes committed by the Russian Federation against the religious communities settled on Ukrainian territory showed that the destruction of churches and religious buildings was both voluntary and deliberate.[8]

The objective of erasing the enemy's "cultural landscapes" as a means of domination, annihilation, and division (Bevan 2006, p. 8) has always been part of war strategies, with the aim of systematic destruction of cultural memory. For instance, even before the term "urbicide" had become part of the international lexicon, the term "coventrialisation", derived from the British city of Coventry—which was completely razed to the ground on the night of 14–15 November 1940 by bombing raids perpetrated by the Luftwaffe,

Nazi Germany's air force—had painfully entered the international lexicon to indicate the destruction of a city, systematically and totally, by aerial bombardment.[9]

The bombing did not spare even the Coventry Cathedral, the city's 14th century landmark, which was hit by 12 firebombs.[10]

The fact that, in the course of the Second World War, there was no special term coined for every location destroyed does not mean that the systematic and deliberate destruction did not affect the cultural heritage of various cities.[11] In February 1945, for instance, when the war was already lost for Germany, the British retaliation to the attack on Coventry followed. The German city of Dresden, capital of the Kingdom of Saxony and known as "Florence on the Elbe" for its monumental and cultural wealth, was razed to the ground by British and US bombing raids.

These attacks demonstrated the substantial ineffectiveness of the then-existing regulatory instruments: the Regulations annexed to the Hague Convention of 1899 and the subsequent Conventions, which were also signed at the Hague in 1907.[12]

The intentional damage or destruction of cultural heritage can be interpreted as a direct attack on a specific human group, so much so as to make the use of the term urbicide inappropriate and to instead warrant the label of genocide, and, more specifically, "cultural genocide". However, there is no trace of this definition of genocide in cultural terms in the 1948 U.N. Convention on the Prevention and Punishment of the Crime of Genocide, which defines the crime of genocide as the physical or biological destruction of a group.[13] The exclusion of cultural genocide[14] was already the subject of strong perplexity at the time, manifested above all in the legal elaborations proposed to the League of Nations before the Second World War by Raphael Lemkin. He, taking a particular interest in the Armenian genocide, wanted to distinguish the physical extermination of an ethnic group (which he called "barbarism") from the destruction of its culture and identity (defined as "vandalism") (Lemkin 1947; Irvin-Erickson 2017, pp. 185, 217). The failure to mention cultural genocide in the Genocide Convention has resulted in its absence from the entire body of international law, since it does not list the systematic destruction of cultural identity as a criminal offence (Bachma 2019, p. 45 ff.). In 1993, the Draft Declaration on the Rights of Indigenous Peoples (UNDRIP), drafted by the Subcommittee on Prevention of Discrimination and Protection of Minorities, in Art. 7, re-proposed and expanded the concept of cultural genocide.[15] However, there was no intention to affect the definition of the "crime of genocide" enshrined in the 1948 Convention; rather, the objective was to outline a broader notion of genocide that would only be effective within the system of protection of the human rights of Indigenous peoples, which a future version of the Declaration would outline. The proposed notion, however, was not accepted in the final text of the Declaration adopted in 2007,[16] in which, with regard to the destruction of cultural heritage, Article 8 (1) proclaims that "Indigenous peoples and individuals have the right not to be subjected to forced assimilation or destruction of their culture". It also indicates, in the following paragraph 8 (2), the type of acts against which States are prepared to adopt effective measures of prevention and compensation. The concept of "forced assimilation" used in the 2007 Declaration, however, in implying the presence of a conflict between a majority and a minority, cannot be applied to the case of the Russian–Ukrainian conflict and, therefore, becomes inapplicable at the International Court of Justice by the Ukrainian government, as the war is between sovereign powers.

The fact that "cultural genocide" is not prohibited in the 1948 Convention nor in international law exposes culture to the continuing risk of targeted attacks that would not constitute crimes against humanity according to the law of war.

In the first months of the Ukrainian conflict, in fact, the world of culture stood by and watched as exceptional sites of history, religion, and art were struck, awakening from its slumber the institutional awareness of the fragility of heritage, for which intervention and protection were repeatedly urged in the world's leading newspapers even before the Russian invasion began.

Four months after the outbreak of the conflict, UNESCO Director General Audrey Azoulay reported the partial or total destruction of 152 cultural and religious sites, including, in addition to the aforementioned Ivankiv History Museum, the Holocaust memorial in Babyn Yar and the theatre in Mariupol.

Following the first bombings in major cities, cultural institutions mobilised staff and volunteers to pack up and transfer museum goods and collections to warehouses prepared to receive them in emergencies. As established by the International Council Of Museums (ICOM),[17] it is the duty of the staff to take pre-established measures for greater effectiveness and coordination in disaster situations,[18] which are not only limited to "physical" damage, but also to the theft and illicit trafficking of cultural property which often occur in conditions of general disorder.

The only comforting news that can be extrapolated from the endless stream of information disseminated from the UNESCO reports is the absence of any reference to damage at the seven recognised World Heritage sites that are not included in the *List of World Heritage in Danger*,[19] but the conflict continues, and culture remains in the crosshairs. Indeed, when it comes to war, culture can never be considered completely out of danger due to its infinite forms of manifestation and local specificities.

The destruction of a site of national importance, or of a site with religious importance, is not only an attack on humanity's treasures, but rather an act of destruction of the social groups settled on that territory. These groups recognise themselves in those cultural and religious sites, since they represent their identity and essence.

Incidentally, while international attention understandably turns (proportionally to the extent of the damage) to the protection of Ukrainian heritage, Russian heritage is also involved in systematic elimination operations, becoming the victim of a social trend which has been recognised in the new millennium: *cancel culture.* Adopted in the world of mass media to describe a "democratised" modernisation of the ancient *damnatio memoriae*, this Anglicism alludes to a form of mass blaming that physically or socially removes the subject accused of "offensive" behaviour. In these circumstances, the target is not an individual, but Russian culture, and, contextually, its traces not only in the post-Soviet Ukrainian cultural heritage and Russian-speaking communities in the easternmost regions bordering the Russian Federation, but also in the western states.[20] On 19 June 2022, the Ukrainian Parliament passed a law aimed at drastically limiting the entry of products from this culture, starting with books and music of post-Soviet Russian citizens, into the country, and a bill[21] is also awaiting approval. This continues the process of total "derussification" that was evidenced by the desire to remove the Arch of Friendship of Peoples in Kyiv (which contains a sculpture depicting two workers, one Russian and the other Ukrainian, which had already been damaged during the first months of the war) and to change the toponymy of certain city streets. Although these initiatives have even been welcomed by some Ukrainian artists, many public figures have spoken out against the idea of boycotting it. One such figure is Sergei Loznitsa, who rebukes his government's stance, firmly arguing that Ukrainian culture is the result of interactions and exchanges with the renegade Russian culture, and its cancellation would unequivocally drag Ukraine's cultural heritage down with it.

It is precisely the Ukrainian director's recognition of the hybrid nature of the two cultures that should motivate any action to protect heritage, with the awareness that it does not belong to a temporary political reality, but to the entire community diachronically. Therefore, in this circumstance of conflict, now more than ever, the two entities of cultural heritage must be protected from the nationalistic logic of war. In this regard, one cannot neglect to mention the role of religion in the conflict. The destruction of the religious cultural heritage in Ukraine reifies the already existing tensions between the two Orthodox Churches, whose definitive schism was announced by the Kyiv Patriarchate following the public advocacy for Russian invasion by the head of the Russian Orthodox Church, Kirill I, who accused the alleged infiltration of Western culture of undermining relations between the two countries. The destruction of places of worship for the Ukrainian population

represents yet another example of "cultural genocide", in which one of the constituent elements of the people's identity is taken away along with the desire to subjugate and assert the supremacy of one political power over the other, ignoring the common and shared roots of the two religious groups that, in the face of war, should ally themselves in order to reaffirm the importance of dialogue and welcome.

The uprooting of a people, a religious group, or a social group inevitably occurs through erasure of the testimonies they have left there. The destruction of material cultural heritage and the consequent systematic mortification of the intangible Ukrainian cultural heritage carried out by the Russian Federation is no less than that which Ukraine is carrying out against the material and intangible products of Russian culture. This constitutes a loss for both cultures involved, as well as very serious damage to the whole of humanity. If, in fact, the erasure of material and immaterial Russian culture in Ukraine leads to problems at the time of post-war reconstruction in defining the identity of the country, or betraying or revisiting its origins, the destruction of Ukrainian material heritage could be the justification for reconstructing it under new assumptions that are based on the cancellation of cultures rather than on a more mature process of growth and searching for one's own identity.

### 3. The Relevant Role of Religion in the Russian–Ukrainian Conflict

Although urbicide is not a recognised category in international law (unlike genocide, but on par with cultural genocide), it represents an important interpretative key to wars. In this paradigm, the destruction of everyday space, cultural symbols, and the population itself are not secondary effects, but part of the strategy of annihilating the enemy. In fact, the term urbicide has come to include not only the damage and destruction of urban historical, artistic, and architectural heritage, but also the destruction directed against a certain way of life, a specific urban culture, a network of relations and activities, or a religion or physical community, which, in the Ukrainian case, has been multi-ethnic for centuries. Different ethnic/religious groups have coexisted on Ukrainian territory since ancient times; in addition to the Ukrainians, which represent the majority, there is a conspicuous Russian minority, as well as smaller Polish, Belarusian, Moldavian, Crimean Tatar, Bulgarian, Hungarian, Romanian, Armenian, Greek, and other minority communities, which are a reminder of the multi-ethnic sedimentation of this borderland. This plural reality is inevitably reflected in the population's religious affiliation. Having dissolved the ancient Jewish community,[22] the majority of the Ukrainian population is religiously divided between Catholicism and Orthodoxy. This division is also reflected at the territorial level, with a western part of the country that is Ukrainian-speaking and traditionally Greek Catholic (Himka 1999) and an eastern part that is Russian-speaking, Orthodox, and pro-Russian in orientation[23]. In addition to the division between Catholicism and Orthodoxy, also complicating the confessional picture is the internal fracture within the Ukrainian Orthodox world. This fracture has led to the birth of other Orthodox currents such as the Ukrainian Autocephalous Orthodox Church (UAOC), born in 1917 and dissolved in 1930, but which survived in the diaspora in the United States, and the Ukrainian Orthodox Church of the Kyiv Patriarchate (UOC-KP), founded at the dawn of the country's independence in 1992 and led from 1995 to 2018 by Filaret (Denysenko). Despite several attempts to create an autocephalous Ukrainian Church, and although Ukraine gained independence from the former Soviet Union in 1991, its only universally recognised national Orthodox Church is still linked to Moscow (Ukrainian Orthodox Church of the Moscow Patriarchate—UOC-MP).

The re-foundation of a Ukrainian Orthodox Church, born in opposition to the Moscow Patriarchate, places the Russian–Ukrainian conflict not only on geopolitical, strategic, and economic grounds, but also on religious ones (Cimbalo 2022). A close alliance between throne and altar has always, in the Russian conception, linked religion to political institutions, whereby religion is seen as an instrument of government and as a vehicle of Russian influence on the world.[24]

The direct involvement of religion and the conflicts between the Orthodox Churches within the context of the ongoing war have led to a number of repercussions on religious cultural heritage. In addition to direct attacks on priests, religious cultural heritage has also been subject to reprisals.[25] This was the case in the village of Lukashivka, where the Orthodox Church of the Ascension of the Lord, a centuries-old architectural landmark and spiritual centre of the community, was transformed into a military headquarters and ammunition depot by the Russian army. In addition, the historic Orthodox monastery "Lavra of the Holy Dormition of the Mother of God" of Sviatohirsk, in the Donetsk *oblast*, was also under the jurisdiction of the Ukrainian Orthodox Church (Moscow Patriarchate) and was destroyed by Russian bombing.

In 2018, the Patriarchal and Synodal Tomos grant of autocephalous ecclesiastical status to the Orthodox Church in Ukraine by the Patriarchate of Constantinople, which sealed the unification of the UOC KP and some parishes of the UOC MP into the Ukrainian Autocephalous Orthodox Church (UAOC), contributed to further straining the relations between the two factions. The intervention of the Patriarchate of Constantinople granted independence to the UAOC, although it has always been denied to the Ukrainian Autocephalous Orthodox Church and the Ukrainian Orthodox Church-Kiev Patriarchate, which have never managed to obtain such recognition (Quintavalle 2018; Parlato 2019).

In the Patriarchal and Synodal Tomos, we read that the UAOC is recognised as the spiritual daughter of the Patriarchate of Constantinople under the name "Most Holy Church of Ukraine", and the canonical territory in which it can extend its jurisdiction is that of the State of Ukraine; it is prohibited to expand into regions already lawfully dependent on the Ecumenical Throne.[26]

The granting of autocephaly by the Ecumenical Patriarch of Constantinople, Bartholomew I, was justified by the fact that the Ecumenical Patriarchate itself declared the 1686 synodal letter by which Dionysius IV placed the Kyiv metropolis under the jurisdiction of the Moscow Patriarchate to be null and void, as it considered it to be of a transitory nature.

This stance was not appreciated by the Patriarchate of Moscow and its Patriarch Kirill I, who saw this gesture as an undue intrusion into his affairs and, in fact, shifted the interests at stake in the ongoing conflict to another level: that of a religious war within a religion between the Patriarchates of Constantinople and Moscow, increasing a rift already evident in the Orthodox Church.[27]

The Ukrainian affair is particularly complex if for no other reason than the spiritual, ecclesial, and cultural heritage that Kyiv and Moscow have shared for centuries. This heritage is, therefore, part of this dispute (Merlo 2019, pp. 194–95), in fact fuelling the ongoing Russian–Ukrainian conflict. The autocephaly granted to the UAOC irretrievably intertwines religious issues with political ones. Although it is not the direct cause of the conflict, it makes it more bitter, especially in terms of an attack on culture, since it threatens to erase the common historical and cultural heritage of the Orthodox communities and lead to a final rupture between Kyiv and the Kremlin.

The city of Kyiv in particular represents the material focus of this political/religious dispute. This is not only because Rus', the oldest form of state of the Eastern Slavs, was born in Kyiv between the 9th and 10th centuries A.D.,[28] but also because the Russian Orthodox Church itself, which traces its historical origins back to Ukraine on the occasion of the baptism of Prince Vladimir I in 988, considers Kyiv to be the spiritual centre of Holy Rus'.

On the other hand, with the collapse of the Soviet Union, Russia found itself needing to build a new identity. Stripped of the territories of the former socialist republics, where millions of Russians suddenly found themselves cut off from their homeland, Moscow relied on the Orthodox Church, guardian of the imperial vestiges with a multinational vocation. Accustomed to keeping and containing different peoples under a single creed, Moscow Orthodoxy became a pillar of the "Russian World", the *Russkij Mir*, a cultural and political project developed in the mid-1990s with the aim of consolidating the interior in order to secure the neighbouring exterior. It is no coincidence that Kirill I and all his predecessors were not patriarchs of Russia alone, but of all the Russias, namely of the

Big one, the White one (Belarus), and the Small one (Ukraine). Separating the latter from the Russian empire means, for Moscow and its Patriarchate, severing what God intended, namely, a single Orthodox community (Luchenko 2023).

The history that binds these two nations, therefore, is particularly intricate, especially from a cultural and religious point of view. The ongoing war has become the occasion for a new concept of urbicide: it is driven by the nationalist spirit that is driving the conflict in reducing cities to rubble, which, at the same time, erases Ukrainian (religious) identity. The bombings, even as the new year begins, continue to target the capital as a demonstration of this intent.

Kirill I's justification that the aggression against Ukraine finds its theological basis in the common membership of Russia and Ukraine in the Orthodox faith, which defend against the Evil One, represented by Western immorality and decadence, provoked the reaction of more than 400 Ukrainian priests of Russian obedience, who demanded that the Patriarch of Moscow be dismissed from his role. Meanwhile, Metropolitan Epifanij, head of the Ukrainian Autocephalous Orthodox Church, and Metropolitan Svyatoslav, head of the Greek Catholic Church of Ukraine, sealed an agreement on 24 December 2022 to change and harmonise their liturgical calendars. Christmas Day 2022 was celebrated on 25 December for both Churches, as opposed to 7 January as the Orthodox tradition dictates. This circumstance, in bringing the Ukrainian Autocephalous Church closer to Western tradition, marked a further departure from the tradition of the Russian Orthodox Church (and from all the other churches in the Orthodox world), since this decision was not an isolated one, but part of a common project intended to lead to the modification of all religious holidays. The aim of this operation was, evidently, for Ukraine to further distance itself from Moscow on a religious level.

This goal is reflected in the adoption of legislative measures to prohibit the Ukrainian Orthodox Church of the Moscow Patriarchate from operating and promoting worship activities throughout the country. This is despite the fact that on 28 May 2022, the UOC-MP Information and Education Department announced the results of the extraordinary convocation of the Council, who examined the issues of Church life that had arisen as a result of the Russian Federation's military aggression against Ukraine. Based on the results of the work, the Council had passed a number of resolutions, including the decision to adopt the appropriate amendments to the Statute on the Administration of the Ukrainian Orthodox Church, which testify to its full independence and autonomy.

However, there is more. A bill submitted to the Ukrainian Parliament, dated 22 March 2022, "On the ban of the Moscow Patriarchate on the territory of Ukraine"[29], as well as the law that entered into force on 26 December 2018, "On amending Article 12 of the Law of Ukraine "On Freedom of Conscience and Religious Organisations" regarding the name of religious organisations (associations) that are part of the structure of a religious organisations (associations), the management center of which is located whitin the borders of Ukraine in a state recognised by law as having carried out military aggression against Ukraine and/or temporarily occupied part of the territory of Ukraine", prohibit the activities of those religious organisations "which directly or as a constituent part of another religious organisation (association) is part of the structure (is part of) a religious organisation (association), the management centre (management) of which is located outside Ukraine in the state, which is recognised by law as having carried out military aggression against Ukraine and/or temporarily occupied part of the territory of Ukraine, is obliged to reflect its affiliation to a religious organisation (association) outside Ukraine in its full name, specified in its statute (regulations), to which it is a part, by mandatory reproduction in its name of the full statutory name of such a religious organisation (association) with the possible addition of the words "in Ukraine" and/or indicating its place in the structure of a foreign religious organisation".[30]

The involvement of religion in the current conflict leads to a series of significant consequences for the country from a political point of view as well. In fact, the measures restricting the activities of the UOC-KP, inevitably affecting freedom of worship, call into

question the country's status as a candidate to join the European Union.[31] The violation of Article 10(1)[32] of the EU Charter of Fundamental Rights should be considered,[33] in addition to the obvious failure to respect the conditions of democracy, the rule of law, human rights, and the protection of minorities (including religious minorities) that the so-called Copenhagen Criteria of 1993 require candidate states to guarantee in order to enter the EU.[34]

## 4. International Instruments for the Protection of Cultural Heritage

"Russia is deliberately destroying Ukrainian culture and our historical heritage. A state that does this cannot be a member of UNESCO and remain at the UN as if nothing had happened". These were the harsh words uttered by Ukrainian President Zelens'kyj while condemning the Russian army's atrocities during the attack on the Dormition Monastery in Svyatogorsk, urging the international community to isolate the oppressor from any inter- and supra-state relations.[35] Since the end of the Second World War, international balances have been maintained by the diplomatic mediation of intergovernmental organisations such as the UN and UNESCO, which have fostered "the incorporation of international clauses and human rights principles" (Scarciglia 2018, p. 281) into the legislation of the adhering countries. These organisations were designed to emphasise the importance of cooperation and the existence of humanitarian objectives that go "beyond the state" within a global legal order.

In order to assess the efficiency of the institutional modus operandi in the current war context, it is necessary to introduce the main legal source of reference in heritage risk situations: the 1954 Hague Convention for the Protection of Cultural Property in the Event of Armed Conflict and its two (1954 and 1999) Protocols. The process that led to the drafting of this Convention was long and articulated; in fact, the concept of the protection of heritage in the event of armed conflict was codified for the first time in the 1907 Hague Convention with respect to the Laws and Customs of War on Land. This came in the wake of Article 8 of the International Declaration concerning the Laws and Customs of War, adopted by the Brussels Conference in 1874. In Art. 27 of the 1907 Convention, there emerged a first timid attempt to recognise special protective measures for culture against possible sieges[36], as it was unable to provide additional artillery strength during world wars.

Precisely because of the conditions in which the historical/artistic heritage found itself in the second half of the 20th century, the newly founded United Nations Educational, Scientific and Cultural Organisation (UNESCO) supervised the drafting of the new 1954 *Hague Convention for the Protection of Cultural Property in the Event of Armed Conflict*, which firstly renewed the use of distinctive symbols, this time adopting a specific one, the "blue shield" (arts. 16–17), as a marker for cultural property, transport, improvised shelters, and the personnel assigned to their protection. Compared to previous conventions that relegated the protection of cultural property, defined only as buildings dedicated to worship, the arts, and the sciences, to a single article shared with hospitals and places of refuge,[37] the 1954 convention constitutes the first example of an international provision dedicated exclusively to providing protection to the dignity of heritage and priority of intervention in war situations. It also prolonged the integrity of cultural property by including procedures to be observed even in times of peace and political stability.

Among the most relevant innovations of the 1954 Hague Convention, it is imperative to also recognise of the existence of a unique heritage of mankind, the fruit of the contribution of all the world's cultures, the spatial positioning of which in the Preamble suggests its universal value with respect to all the provisions listed. Preaching the common cultural root of mankind after decades of nationalistic ideologies and racial segregation symbolises the will to separate culture from its function of legitimising the political class. It also prefigures the recognition of the phenomenon of "cultural hybridisation", whereby no culture is isolated, but the result of exchanges, acquisitions, and transmissions between different groups. The acknowledgement of this observation by the signatory states and UNESCO translates theoretically into a duty to ensure the spread of a pluralism of values

after centuries of Western hegemony, in addition to the assumption of responsibility for protection, irrespective of the geographical and cultural context. The vindictive attitude already held towards Russian culture in Ukraine notwithstanding, the contribution of Western countries to the preservation operations remains considerable. Italy has offered to rebuild the Mariupol theatre[38] and the Smithsonian Cultural Rescue Initiative,[39] which has established direct communication networks with Ukrainian cultural institutions to monitor the artifacts and possibly send packing materials for security. In addition to the governmental aid which is often discussed, there is a further network of humanitarian aid with less media visibility, but a greater impact on the population with whom it works: cultural NGOs, the symbol par excellence of heritage protection that transcends the political and cultural borders of states and acts, promptly without the impact of the long regulatory and bureaucratic timeframes required by national laws.

Regardless of the reassuring results regarding civil–military cooperation, one must, in fact, recognise the criticalities of the international decision-making apparatuses that have emerged in this emergency situation and that stem from the very limits of legal globalisation.

Despite the fact that we are in the presence of international regulations that are inadequate to guarantee an effective and concrete system of protection for cultural assets, in the international context there is always greater attention being paid to the formal and content-related evolution of art. This is reified in the issuing of protective measures involving new artistic forms, such as the Conventions safeguarding intangible heritage (i.e., artistic manifestations and expressions that cannot be traced back to material supports)[40] and cultural expressions (i.e., the complex of cultural phenomena that diversifies the global entity of human heritage),[41] but the issue becomes much more delicate in legal matters. Global law is mainly expressed by means of treaties and soft-law instruments, delegating to individual states the tasks of legislating on the subject and preparing for non-compliance by means of sanctions with long-term and democratically planned effects. Therefore, international organisations cannot impose themselves upon the sovereignty of a state; they can only issue resolutions voted upon by a majority. In the specific case of the Russian–Ukrainian conflict, the UN General Assembly managed to expel Russia from the Human Rights Council, but cannot expel it from the organisation itself as Zelens'kyj would prefer. This is because, in accordance with Art. 5 and 6 of the United Nations Charter,[42] suspension or expulsion measures are a prerogative of the Security Council, in which Russia sits among the permanent member states. The prospect of expelling Russia from this decision-making council can be ruled out, since any changes to the body would have to be voted upon unanimously by the Council itself.

Similarly, regarding the preservation of cultural property, UNESCO does not have the power to intervene on its own initiative, neither from a sanctioning nor a military point of view, but it has proven capable of actively contributing to the partial restoration of a sense of collective security. In the preceding months, UNESCO committed to promoting integration projects for the education of Ukrainian students abroad and international students residing in Ukraine,[43] and worked with the Ukrainian authorities to mark cultural sites with the "Blue Shield" symbol required by the 1954 Convention. In addition, on 1 July 2022, the Evaluation Body for Intangible Heritage of Humanity nominations included Ukrainian *borscht* in the UNESCO intangible heritage list,[44] a typical dish made from *borchevik* (Caucasian hogweed, Heracleum sphondylium, a herbaceous plant that grows in moist grasslands near the Danube and Dnieper rivers). The recipe was disputed with Russia, which claimed paternity. As much as the media interpretation of this decision attributes it to a political stance on the part of the international body, it is sufficient to read the text of the 2003 Convention for the Safeguarding of Intangible Cultural Heritage[45] to refute the widespread geopolitical distortion and understand the main purpose for which the analysis of this candidacy was urgently anticipated. Inclusion on the list ensures the preservation of practices and traditions while respecting cultural diversity regardless of the country proposing the candidacy; the recognition of the dish as a World Heritage Site does not attribute authorship to Ukraine, but favours its intergenerational transmission

and protection. In relation to this last point, however, it is necessary to clarify that a UNESCO listing does not guarantee a special or unique system of protection. Therefore, the recent proposal of candidacy for the World Heritage in Danger list for the historic centre of Odessa[46] represents an opportunity for Ukraine to focus international attention on that site, but in the limited and unsuccessful forms which, over the years, have consigned numerous treasures of human civilisation into the hands of terrorists and warlike destruction while awaiting the normative evolution of the existing Conventions.

To offset the positive results of UNESCO's interventions in cultural matters, certain flaws in the wording of the procedural provisions on the protection of cultural property during conflicts remain evident.[47]

There are numerous citable cases of destruction of cultural property after the Convention, including the famous Mostar bridge, a victim of the Bosnian–Croat war, as a symbol of the union between the Christian and Muslim communities in southern Bosnia and Herzegovina. A further factor limiting UNESCO's actions is the provision of peacetime procedures, which are particularly fragile in terms of their impact on national legislation. What can be deduced by consulting Articles 3 and 7 of the 1954 Convention is the decision to leave it to individual signatory states to adopt specific measures, creating heterogeneous protection systems that are difficult to harmonise in situations of cooperation or exchange between cultural institutions.[48]

In an attempt to remedy the definitional uncertainty and ineffectiveness of the 1954 Convention, the enactment of the Second Additional Protocol in 1999 established a system of enhanced protection, in place of special protection, and a list of general protective measures to be observed by all signatory states. Regarding the first novelty, Chapter 3 is dedicated to more clearly setting out the importance of giving cultural property immunity from deliberate attack or damage under certain conditions, according to short procedures supervised by a specific and competent body, namely, the Committee for the Protection of Cultural Property in the Event of Armed Conflict.[49] The other novelty is the presence of guidelines to be followed in peacetime, which, while remaining general in nature, constitute a first step towards a common concept of continuous protection (Art. 5). The partial amendment of the Convention, unfortunately, did not prevent the cultural massacres caused by the 2001 destruction of the two giant Buddhas in Bamiyan (Afghanistan), nor the looting of the Iraq Museum in Baghdad in 2003, nor did it hinder Russian forces when they chose to systematically attack Ukrainian cultural sites. These circumstances serve to highlight a further weakness that reiterates the deficient nature of the international protection system: the lack of a sanctioning apparatus to be applied against national political entities in the event of non-compliance. After a careful reading of Chapter 4 of the Second Protocol, entitled "Criminal responsibility and jurisdiction", it can be stated that violations of the Second Protocol only concern the individual.[50] No responsibility for wrongful acts arises on the part of an offending state, a concept that is also reiterated in Article 38, which exempts nations from any obligation to repair the damage.[51] One can enumerate the few instances in which provisions on the protection of cultural property took on an internationally binding character, including with reference to Resolution 2253 (2015)[52], adopted by the Security Council following the destruction of Iraqi and Syrian cultural heritage, which was committed in particular by IS and the Al-Nusra Front. This measure included a warning for UN states to take appropriate mandatory measures to prevent the trade of Iraqi and Syrian cultural property and objects of archaeological, historical, cultural, scientific, or religious value illegally removed from Iraq, as of 6 August 1990, and Syria, as of 15 March 2011, by prohibiting their transnational trade and thus allowing their return to the Iraqi and Syrian peoples.[53] The binding character could be applied to the provisions because the Council invoked Article 39 of the UN Charter, which allows it to make autonomous decisions in order to ensure international peace and security in the event of an established presence of a threat or violation of these ordinary conditions.[54] However, these were circumstantial measures, limited to the conflicts in Iraq and Syria,

"while general experience shows that atrocities committed against cultural heritage are a generalised phenomenon in all armed conflicts" (Urbinati 2019, p. 89). Turning again to the current situation, the impediments regarding the involvement of the Security Council in the case of the Russian–Ukrainian conflict have already been underlined, reaffirming the precarious condition of stability in which culture, targeted by both belligerent powers, finds itself.

While the heterogeneity of the procedures and institutions involved evokes a strong feeling of solidarity and cooperation, it is also indicative of the lack of systematic and internationally regulated reference provisions. Once again, what emerges is a poorly understood dialogue between art and law that seeks, in the little time available, to remedy the mistakes of the past.

*New Solutions and Old Problems*

So far, an overview of the condition of Ukraine's cultural heritage has been outlined in an account of all protection procedures put in place to ensure the preservation of the country's history of identity. The implementation of the Hague Protocols for the "physical" protection of cultural property in situ, international aid, and UNESCO interventions was examined. To conclude the discourse on the forms of heritage protection in this war context, one cannot avoid investigating the role that non-fungible tokens (NFT) are playing in the fight against Russian colonialist policies towards Ukrainian culture. The exploitation of this revolutionary technology as a crowdfunding method boasts exponential growth since the initiative of a digital artists' organisation, the decentralised autonomous organisation Ukraine DAO,[55] which auctioned the Ukrainian flag in NFT format (Simeone 2022) online. After its commercial success (it sold on 2 March for ETH 2250, about EUR 6 million), the Ukrainian government opened a web portal[56] offering the possibility to buy NFT digital artworks for the purpose of financing army activities and civil rescue operations.

In the wake of these initial crypto-art exploitation initiatives, on 25 March 2022, Ukrainian Minister of Digital Transformation Mychajlo Fedorov announced via Twitter the launch of the "MetaHistory: Museum of War" project,[57] which provided the opportunity to purchase NFTs of unpublished works retracing key moments of the Russian–Ukrainian conflict between 24 February and 30 April 2022. The 459 available tokens consist of tweets on daily commentary and news about the destructive events and ongoing diplomatic strategies, accompanied by creative digital contributions from Ukrainian and international artists. For the technical realisation of the project, the Ukrainian government turned to the *Ethereum* platform, which took on the task of converting the artworks into NFTs,[58] providing a secure and reliable transaction system for donors and ensuring the devolution of the total sum gained from the sales to the ministry offices. On the other hand, from a social point of view, the objective reported by the government was to document the events of the war as truthfully and in as much detail as possible in order to avoid any distortion resulting from the Russian propaganda monopoly by using the tools of public opinion and the international network of artists supporting the restoration of order and peace.[59]

The Ukrainian government's project is not the first example of an NFT museum, but follows in the wake of projects that have musealised extremely recent artistic trends, even with regard to the very concept of contemporary art. Despite the dizzying attention paid to such initiatives, legislation still proves to be very weak in defining a clear regulatory framework for this new, bivalent form of protection for artists who wish to authenticate their works for the fruition of an increasingly "digital" public. The legal nature of NFTs is not institutionally defined and does not fall into any category of intended work support; they do not constitute works of art, although their entry into museum facilities implies their inclusion in the cultural heritage[60] and, thus, the inherent need for reforms in content and regulatory language. However, the conservative attitudes of institutions often manifest in hasty interventions, a symptom of a lack of understanding and irrational fear of losing the artistic tradition rooted in national cultures.[61]

Against this background, therefore, two novel issues emerge for the art world for which some explanatory factors need to be provided. On the one hand, there is a revolutionary change occurring in the world of collecting; it will no longer concern the physical possession of the work purchased, since NFTs do not involve the purchase of the work itself, but rather a certificate of authenticity attesting to the unique and unrepeatable existence of that transaction. As can be seen from Hirst's project/experiment The Currency (Roccella 2022), the value of art is progressively shifting from a level of historical/artistic interest to a utilitarian and speculative one. The very condition whereby NFTs become objects of interest for contemporary collectors demonstrates a frenetic incorporation of art and culture into the world of the financial market, creating the conditions for the emergence of speculative bubbles, computer fraud, and the circulation of fake works. On the other hand, extending the issue of the commercialisation of works to the institutional sphere, receiving donations and funding through the sale of NFT works of art can be interpreted as the most concrete manifestation of the total alienability of heritage, whereby culture and art risk being sold off in periods of financial and economic crisis through privatisation and monetisation procedures (Settis 2002). The invisible threat of heritage dispersion is facilitated by the very technology of NFTs, which do not imply the transfer of ownership to the buyer of the token and, therefore, do not violate the legal constraints linked to the public dimension of cultural property. This conclusion leads to the exclusion of NFT sales from the case of heritage alienation and, consequently, allows for the proliferation of projects in favour of their integration into museum policies. Moreover, the possession of an NFT accompanied by a copy of the work cannot represent a risk of conservation negligence on the part of the purchaser, which is the parameter the State uses to apply the right of pre-emption and prohibit its circulation; therefore, its dissemination cannot be systematically hindered except by a revision of the policies on the marketing and reproduction of digital cultural assets. The sale of art also concerns the situation of Ukrainian heritage, which is virtually fragmenting in the hands of millions of users of cryptocurrency platforms seeking, in art, the economic solution to the end of the conflict.

It can, therefore, be said that art has always been a tool for communicating wars, massacres, and patriotic deaths thanks to its visual and expressive power (from Trajan's Column to Picasso's Guernica to Yugoslavian memorials, to name but a few examples), but never has it been so intensively involved in supporting the economy, public information, and national identity as during this conflict, especially by the Ukrainian government. Artistic production has become an alternative resource for receiving donations, a tool for visual knowledge of the events of the war, and an invisible weapon of cultural defence which, combined with reproduction techniques, boasts a speed of dissemination equal to that of the news circulating on the web and on social networks.

The perception of the horrors and destruction caused by war underwent a radical change in the second half of the 20th century, after the most recent world war lacerated the whole of humanity. Post-war societies sought, in the new artistic means of communication (photography and cinema), the appropriate tools to represent those events by arousing universal, shared emotions. The spectacularisation of the event (Cati 2016, p. 57) is the expression that best describes the obsession of contemporaries to document any social aspect potentially useful in shaping collective memory through visual and emotional representation. Paradoxically, however, their continuous reworking distances the media from the event itself, and, consequently, negates the very objective for which it was made in the first place. The distorted and repeated information generates a weakening of public interest, a blurred perception of the seriousness of the facts, and, finally, the normalisation of the tragedy by continuous public exposure that "trivialises" and nullifies the singularity of the event (Benjamin 1966, p. 23). War is also deprived of its elements of exceptionality and unrepeatability due to its contact with technical and digital reproduction techniques. In the specific case of the Russian–Ukrainian conflict, it is pointed out that the symptoms of *anaesthetisation* were already evident only two months after the rise of hostilities fuelled by

bombardment from every media platform, from news bulletins to TikTok to the emerging NFT market (Sorice 2022).

The framework presented so far allows us to expose a further problem relating to the breadth of phenomena encompassed by the term "art", as applied in the international legal sphere. It has already emerged above how much the definitional taxation of state legal systems excludes manifestations of contemporary art from protection systems, but a similar problem arises when the law adopts excessively vague concepts that risk being subverted and manipulated. The lack of an unambiguous legal definition of the word "art" allows it to be applied to any human product as soon as it is declared as such by the artist or the community, further complicating the attribution of objects of protection and opening the door for unrestrained exploitation of the communicative potential of art. In the case of the *MetaHistory: Museum of War* project, the desire for possession of the *tokens* unequivocally ended up converting art into a tool for the commercialisation of war and the spectacularisation of tragedy, placing itself in open contrast to the initial objectives set by the government.

Amidst the explosion of NFT sales by Ukrainian cultural productions and the phenomenon of deaccessioning[62] of museums, protecting art at this historic moment also means rescuing it from the commercial abuses of which it has become a protagonist, and entails the realisation that the danger of losing its historical and artistic value lurks even within those spaces deemed "safe" for its preservation.

## 5. Conclusions

War not only has the power to erase memory in the immediate term, but also acts in the long term. Even the work to reconstruct what has been lost or destroyed can contribute to memory erasure, especially in the presence of attempts to manipulate or otherwise revisit collective memory (Mazzucchelli 2010).

The UNESCO-sponsored reconstruction of the bridge in Stari Most, or the future rebuilding of the theatre in Mariupol, subsidised by Italy, are only two of the many examples that could be cited to concretely summarise the practice of post-conflict rehabilitation of immovable cultural properties. Unfortunately, not all buildings, however culturally relevant they may be in the broadest sense, can benefit from this type of international aid, not only in terms of economic funding, but also in terms of guarantees. In fact, the direct intervention of impartial third parties in a country's reconstruction process can act as a guarantee for the preservation of historical memory. Although the digitisation of cultural heritage may help to keep tradition alive,[63] reconstruction, where a form of control is lacking, could be accompanied by a process of historical and cultural revisionism that, especially with regard to places of worship, could lead to serious limitations on the fundamental right of religious freedom and religious pluralism.

Ukraine's pluralism has always been reflected by its urban centres, and Kyiv is a clear example of this. The urbicide that has been affecting the city in recent weeks has become the precise objective of the military strategy, guided by nationalist logic: to alter the urban space of Kyiv, which has always housed the buildings, symbols, and institutions dear to both Russian and Ukrainian Orthodoxy. With the autocephaly granted by the Ecumenical Patriarchate to the Ukrainian Orthodox Church, Kyiv's cultural heritage is no longer perceived as a symbol of the richness of a society in which Russian and Ukrainian cultures coexist, but, on the contrary, becomes a symbol of the historical presence of the "other". Thus, the urban landscape begins to be viewed through the lens of nationalist politics and cancel culture: the sight of an Orthodox church of the UOC MP or the Ukrainian autocephalous church is interpreted not as a positive consequence of pluralism, but as proximity to the enemy.

For this reason, in the reconstruction process, places of worship should be among the first buildings to be renovated or reconstructed, as they are ethnic and religious symbols that serve to create specific spatial landmarks in the city; this will aid in initiating the process of reconciliation between Russian and Ukrainian cultures.

Although, at first glance, the temptation of Ukraine to eradicate all traces of Russian cultural and religious influence from its territory (cancel culture) might be understandable, such behaviour would lead the country to transition from aggressed to aggressor by fomenting further rifts within the population. In order to prevent Ukraine from becoming embroiled in a new endogenous conflict, care must be taken to ensure that memory and tradition are guaranteed in the course of reconstruction, because change and the search for identity must be the result of a process desired by the population and not imposed from above. In the face of international conventions which are inadequate to protect cultural heritage in the event of war, and the activities of UNESCO, which are not fully effective in intervening for its protection, the case of Ukraine highlights a clear need to recognise heritage as an active subject of law, to place constraints on the exploitation of art when it undermines the transmission of its humanitarian values, to guarantee religious pluralism, and to use cultural heritage as a form of resistance to Ukrainian and Russian memory-removal policies.

**Author Contributions:** Sections 1–3 and 5 written by F.B. and Section 4 by C.B. All authors have read and agreed to the published version of the manuscript.

**Funding:** This research received no external funding.

**Conflicts of Interest:** The authors declare no conflict of interest.

## Notes

1     In 1589, the patriarch Jeremias II of Constantinople, with an assembly of bishops, presided over the enthronement of the first Russian patriarch of Moscow and all of Russia, metropolitan Job, marking the beginning of the autocephalous status of the Russian Orthodox Church. Before the establishment of the patriarchate, in fact, the Russian Church was led by the metropolitan. Until the mid-fifteenth century, it belonged to the Patriarchate of Constantinople and had no independent government; it was only after the fall of Byzantium that the Metropolitan of Moscow obtained independence from the Patriarchate of Constantinople Church. In 1721, during the reign of Peter I, the patriarchate was abolished and the emperor established a theological council, later renamed into the Holy Synod—the state body of the highest authority of the Church—and in 1917, according to a decision of the All-Russian Local Council, the patriarchate was restored. (https://www.britannica.com/topic/Russian-Orthodox-Church. Accessed on 20 December 2022). The current Moscow Patriarchy was created in 1943 by Stalin when, in his office in the Kremlin on the night between 4 and 5 September 1943, he received the Metropolitan of Moscow and Kolomna Sergij—locum tenens of the patriarchal throne—the Metropolitan of Leningrad and Novgorod Aleksij and the Metropolitan of Kiev and Galic Nikolaj. All this took place in the presence of Molotov, the head of the NKGB Merkulov, and Colonel Georgij G. Karpov, head of the fifth department of the second directorate of the NKGB, whose competencies included the control and repression of religious organisations (Roccucci 2011, p. 174).

2     https://www.theguardian.com/world/2022/mar/13/russian-orthodox-church-in-amsterdam-announces-split-with-moscow (accessed on 20 December 2022).

3     In this context, expressions of culture are understood as those encapsulated in the concept of "cultural heritage", which encompasses not only the tangible manifestations of man's creative genius (i.e., movable and immovable property), but also the expressions of peoples' intangible cultural identity, i.e., the practices, knowledge, traditions, and know-how that groups and communities (and sometimes individuals) recognise as part of their memory and heritage (Frigo 2004, pp. 367–68).

4     Urbicide was first heard of in the 1990s during the war involving the territories of the former Yugoslavia. The term, from the Latin urbs (city) + caedere (to demolish, to kill), literally means violence against cities. The war conflict at the time brought attention to the phenomenon of material, cultural, and urban identity destruction. On the genesis of the concept of urbicide (Coward 2004, pp. 154–71; Graham 2004): "Urbicide is the liturgical murder of the city, a premeditated and ordered one, with an explicit form. It is the result of actions that wipe out systems of common life's meaningful places (squares, monuments, libraries—the agora), ravage the city's material basis (infrastructure, services—the "urbs"), exterminate society and citizenship (the civitas), and annihilate institutional marks of the government (privatisation, deregulation, centralisation—the "polis")". This type of murder arises in diverse situations, which fit into three types: natural (when it is caused by an aggression of nature, such as a hurricane, fire, earthquake, eruption, or drought.), anthropic (when it is the result of entirely anthropic reasons, from military conflagrations to real estate speculation), or symbolic (e.g., by changing the name of a city, one kills the past and marks a new possession or domain) (Carrión Mena 2018, p. 5). The ferocity of the ongoing war is threatening to erase the cities and cultural history of the Ukrainian people. Also in danger are monuments that are part of the World Heritage of Humanity, such as the Cathedral of St. Sophia in Kyiv, the medieval old town of Lwów, and the Potemkin Stairs in Odessa.

5     Of the 24 *oblasts* (regions) into which Ukrainian territory is divided and their respective capitals, almost half have been bombed. These include Kyiv, bombed by several Russian missile attacks; the eastern city of Zaporizhzhia, home to the largest nuclear

power plant in Europe; Dnipro, also in eastern Ukraine; and the port city of Mykolaiv in the south. The city of Kharkiv and its *oblast* were also bombed; and the city of Trostyanets, in the Sumy region, was liberated by Russian occupation troops, but only rubble remains of it. The city of Mariupol (located in the Donec'k *oblast* and capital of the district of the same name, now part of the Donec'k People's Republic) was largely destroyed after weeks of shelling. Also under missile attack was Nizhyn of the Černihiv *oblast* and its capital of the same name. Zhytomyr, Ternopil, and Lwów, on the Polish border, were the object of a blind retaliation aimed not only at hitting strategic targets, but at sowing terror and devastation among the civilian population.

6    The monitoring was conducted by the State Service of Ukraine for Ethnopolitics and Freedom of Conscience (DESS) in cooperation with the Workshop for the Academic Study of Religion. https://risu.ua/en/ancient-orthodox-church-of-ukraine-legally-withdraws-from-its-subordination-to-moscow_n133680 (accessed on 20 December 2022).

7    See *below* in the text.

8    These are the results of the "Religion in Fire" project developed by the academic community of religious studies in Kyiv and supported by DESS and the Congress of National Communities of Ukraine. https://risu.ua/en/the-researchers-conclude-that-the-russian-military-often-destroys-churches-and-religious-buildings-on-purpose_n132131 (accessed on 20 December 2022).

9    https://www.wordsense.eu/coventrate/ Accessed on 21 December 2022.

10   The air attack on Coventry cost 1236 lives and injured thousands; 4330 homes were destroyed, along with 2 hospitals, 3 churches, 80 per cent of factories, air raid shelters, railway and police stations, post offices, cinemas and theatres, the entire tram and road transport network, power stations, and gas and water distribution networks.

11   The devastation of important cultural sites in the various contexts of warfare are today's evidence of a real strategy which dates far back in time (the words of Cato the Elder, for example, that Cartago delenda est still echo); but it is also true that, on other occasions, such destructions were justified by requirements of a strictly military nature. Speaking in today's terms, these campaigns responded (or so it was believed) to the criteria of necessity and (military) advantages that later found their regulation both in the Geneva Convention of 1949 and in that signed in The Hague on 14 May 1954, the object of which was precisely the protection of cultural property in the event of armed conflict.

12   Many provisions of The Hague Conventions of 1899 and 1907 can be traced back to the Project of an International Declaration concerning the Laws and Customs of War of 1874, a non-binding document drafted at the Brussels Conference. It is relevant since it constituted a first step forward in the codification of the laws of war (McGeorge 2006, p. 209). Article 8 of the Declaration of 1874 offers an early example—in embryonic form—of the protection of cultural heritage in international law. According to the article, the seizure, destruction, and wilful damage of "The property of municipalities, that of institutions dedicated to religion, charity and education, the arts and sciences even when State property, shall be treated as private property. All seizure or destruction of, or wilful damage to, institutions of this character, historic monuments, works of art and science should be made the subject of legal proceedings by the competent authorities". Art. 17 emphasised that: "In such cases all necessary steps must be taken to spare, as far as possible, buildings dedicated to art, science, or charitable purposes, hospitals, and places where the sick and wounded are collected provided they are not being used at the time for military purposes. It is the duty of the besieged to indicate the presence of such buildings by distinctive and visible signs to be communicated to the enemy beforehand". The Declaration contained rules for siege and bombardment aimed at sparing hospitals and buildings of cultural, scientific, religious, or other social importance to the greatest degree possible, as well as an order to prevent looting (McGeorge 2006, p. 204 ff.). The text of the Declaration can be accessed from the International Committee of the Red Cross website: www.icrc.org (accessed on 27 December 2022).

13   Convention on the Prevention and Punishment of the Crime of Genocide (Genocide Convention) was the first human rights treaty adopted by the General Assembly of the United Nations on 9 December 1948. It signified the international community's commitment to "never again" after the atrocities committed during the Second World War. "Article II. In the present Convention, genocide means any of the following acts committed with intent to destroy, in whole or in part, a national, ethnical, racial or religious group, as such: (a) Killing members of the group; (b) Causing serious bodily or mental harm to members of the group; (c) Deliberately inflicting on the group conditions of life calculated to bring about its physical destruction in whole or in part; (d) Imposing measures intended to prevent births within the group; (e) Forcibly transferring children of the group to another group".

14   It is emphasised that the introduction of the notion of "cultural genocide" was proposed in the *draft* Convention prepared by the UN Secretary General, the *Draft Convention on the Crime of Genocide* of 26 June 1947, in which Article 1(3) included a mention of genocidal acts: "[d]estroying the specific characteristics of the group by: (a) forcible transfer of children to another human group; (b) forced and systematic exile of individuals representing the culture of a group; (c) prohibition of the use of the national language even in private intercourse; (d) systematic destruction of books printed in the national language or of religious works or prohibition of new publications; (e) systematic destruction of historical or religious monuments or their diversion to alien uses, destruction or dispersion of documents and objects of historical, artistic, or religious value and of objects used in religious worship".

15   Article 7: "Indigenous peoples have the collective and individual right not to be subjected to ethnocide and cultural genocide, including prevention of and redress for: (a) Any action which has the aim or effect of depriving them of their integrity as distinct peoples, or of their cultural values or ethnic identities; (b) Any action which has the aim or effect of dispossessing them of their lands, territories or resources; (c) Any form of population transfer which has the aim or effect of violating or undermining any

of their rights; (d) Any form of assimilation or integration by other cultures or ways of life imposed on them by legislative, administrative or other measures; (e) Any form of propaganda directed against them".

16    The United Nations Declaration on the Rights of Indigenous Peoples (UNDRIP) was adopted by the General Assembly on Thursday, 13 September, 2007, by a majority of 143 states in favour, 4 votes against (Australia, Canada, New Zealand, and the United States), and 11 abstentions (these include the Russian Federation and Ukraine).

17    The International Council of Museums (ICOM) is a non-governmental organisation dedicated to museums. It maintains formal relations with UNESCO and has a consultative status with the United Nations Economic and Social Council.

18    ICOM Code of Ethics for Museums, paragraph 1.6, *Protection* Against Disasters, and paragraph 2.21, Protection Against Disasters. https://icom.museum/wp-content/uploads/2018/07/ICOM-code-En-web.pdf (accessed on 20 December 2022).

19    Convention concerning the Protection of the World Cultural and Natural Heritage adopted on 16 November 1972, art. 11, c. 4 "[ . . . ] The list may include only such property forming part of the cultural and natural heritage as is threatened by serious and specific dangers, such as the threat of disappearance caused by accelerated deterioration, large- scale public or private projects or rapid urban or tourist development projects; destruction caused by changes in the use or ownership of the land; major alterations due to unknown causes; abandonment for any reason whatsoever; the outbreak or the threat of an armed conflict; calamities and cataclysms; serious fires, earthquakes, landslides; volcanic eruptions; changes in water level, floods and tidal waves. [ . . . ]".

20    The damage to cultural heritage since the start of the conflict has been transversal in that it has not only affected Ukraine, but also, reflexively, Russia. In fact, one of the many side effects of the Russian invasion has been the reaction by Western countries to strike at the products of Russian culture. In Italy, the most famous case was that of the writer Paolo Nori, who had to give up lectures on Dostoevsky at the Bicocca University in Milan. Also, at the University of Leeds in England, the editors of the journal *Studies in the History of Philosophy* decided to forego a thematic issue on Russian religious philosophy, since it was feared that the works of Russian religious thinkers of the 19th and 20th centuries could be used for propaganda purposes. But there have also been cases on other sides of culture: the *Children Book Fair* in Bologna (Italy), for example, suspended all cooperation with Russian organisations; and the *Galleria Accademia* in Florence and the *Royal Opera House* in London were also banned from featuring Russian artists. In Italy, the *European Photography Festival,* which had Russia as a guest, was cancelled because "the organisers cannot have relations with a country that is an aggressor" (https://artslife.com Accessed on 18 December 2022). The cancellation *culture* also affected gastronomy; again, in Italy, the attempt to boycott the "Russian salad" ended up turning into a *butade,* since in Moscow and the rest of the world, they know that dish as "Olivier salad". But Italy was not the only one to attempt to censor the food industry: abroad, Moscow Mules have been renamed Kyiv Mules. Brighton Beach's grocery store, Taste of Russia, in Brooklyn, changed its name, as did Washington, D.C.'s Russia House in Dupont Circle (https://reason-com. Accessed on 18 December 2022). To date, the work of erasing Russian culture, at least in Italy, seems to have come to an end. For example, the premiere at La Scala in Milan opened its 2022–2023 season with Modest Petrovič Musorgskij's *Boris Godunov* (https://www.teatroallascala.org/en/index.html (accessed on 18 December 2022).

21    This is the "Draft Law on Amendments to Certain Laws of Ukraine Establishing Restrictions on the Importation and Distribution of Publishing Products Concerning the Aggressor State, the Republic of Belarus, the Temporarily Occupied Territory of Ukraine", dated 13 June 2022, as yet unsigned by the Supreme Council of Ukraine (*Verchovna Rada* of Ukraine; Ukrainian: Верховна Рада України). It provides for the amendment of the Law of Ukraine "On Publishing", No. 32 of 1997 (Information of the Verkhovna Rada of Ukraine–Відомості Верховної Ради України (ВВР), 1997, No. 32, ст.206) and prohibits the printing of books produced by authors who, after the dissolution of the Soviet Union in 1991, retained Russian citizenship, unless they renounced it and took Ukrainian citizenship. The import of Russian books printed in Russia, Belarus, or the "temporarily occupied Ukrainian territories" is prohibited. https://itd.rada.gov.ua/billInfo/Bills/Card/39764 (accessed on 16 December 2022). The "Draft Law on Amendments to Certain Laws of Ukraine Concerning Support for the National Music Product and Restricting Public Use of the Music Product of the Aggressor State" of 30 May 2022 completed its processing on 7 June 2022. It introduces amendments to the following two laws: the Law of Ukraine "On Culture", No. 24 of 2011 (Відомості Верховної Ради України (ВВР), 2011, No. 24, ст.168); and the Law of Ukraine "On Ensuring the Functioning of the Ukrainian Language as a State Language", No. 21 of 2019 (Відомості Верховної Ради (ВВР), 2019, No. 21, ст.81). As a result of these amendments, the reproduction of music by post-Soviet Russian artists is prohibited on Ukrainian media and public transport, with the aim of increasing Ukrainian-language programmes and music on radio and TV. Excluded from the ban are musicians who condemned Moscow's invasion, who will be included in a special "white list". https://itd.rada.gov.ua/billInfo/Bills/Card/39702 (accessed on 16 December 2022). These laws and drafts are just the latest steps in a long journey that Ukraine has been on for several years now to overcome the legacy of Soviet domination and regain its national identity, called, until recently, *decommunisation.* After having been intensified since 2014 following Russia's invasion of Crimea and the start of the conflict in the Donbass, today, this process has gained more and more ground, and no one is afraid to speak explicitly about *derussification* anymore. In 2019, for example, a law was passed requiring civil servants to know Ukrainian. Since the outbreak of the war, however, hundreds of places in Kiev alone have decided to change their names to remove any reference to their links with Russia, and a Soviet-era statue erected to celebrate the friendship between the two countries has been torn down. On the other hand, it must be remembered that measures of this kind are not new in the political history between the two countries. On the Russian side, a policy defined as "attacking the Ukrainian language" began in 1654 with the annexation of Ukrainian territories to Russian rule. Russification reached its peak with the issuing of two decrees by the Tsar in 1863 and 1872, respectively. The first, called the Valuev Decree, stated that "the majority of the Little

Russians (the name of Ukrainians in the Russian empire) prove that any Ukrainain language ever existed, exists or will exist". It was "the same Russian language with the only difference of it being spoiled by Polish"; and again, that "the Russian language is understood by all the Little Russians far better than their Little Russian (that is Ukrainian) language", and that "Some poets call this language Ukrainian". The second decree (Emsky decree) went so far as to prohibit the import of books in Ukrainian if they had been published in western Ukraine (Totskyi 2010).

22　At the end of the 19th century, there were more than 2 million Jews in Ukraine. However, after the exiles and murders of the Bolshevik revolution of 1917 and then the invasion of Nazi Germany in 1941, only 800,000 Jews remained in Ukraine after World War II. Before the German invasion, about 160,000 Jews resided in Kyiv, or about 20% of the capital's total population. At the time of the German occupation of Kyiv, about 60,000 Jews remained in the city, more than half of whom were massacred between 1941, remembered today as the Babyn Yar massacre, and 1943. The Jewish presence was a constant in the Kyiv area (Nathans 2002), so much so that, despite the *pogroms*, Kyiv was a multi-religious and multi-lingual city where Jews and other national groups (Ukrainians, Russians, and Poles) interacted in a context that we would call intercultural today (Meir 2006, p. 485). Despite the fact that with the dissolution of the USSR in the 1990s, a large component of the Jewish community emigrated to Israel, in Ukraine, the Russian-speaking Israeli community was the third largest in the country. To date, following another major emigration that took place during the conflict that erupted in the Donbass in 2014, the country's Jewish component has shrunk further.

23　See: U.S. Department of State, Ukraine 2021 International Religious Freedom Report, Section I. Religious Demography, p. 4, https://www.state.gov/wp-content/uploads/2022/04/UKRAINE-2021-INTERNATIONAL-RELIGIOUS-FREEDOM-REPORT.pdf (accessed on 2 January 2023).

24　Demonstrating the existence of this symphonic relationship, the close relationship between the two institutions was sealed in 2011 when the Moscow Patriarch's residence was moved inside the Kremlin, the seat of political power.

25　On 1 January 2023, the lease by the UOC MP of the Dormition Cathedral and the Church of the Feast of Kiev Pechersk Lavra, a historic Kiev monastery also known as the Cave Monastery, expired. These buildings, which have always been in the possession of the UOC MP, have been seized by the state. A commission composed of representatives from the National Nature Reserve, the Miller Law Company, the UOC MP, and the Culture Ministry carried out the transfer, which included taking inventory of the property and preparing a technical inspection report for the two buildings. https://www.pravda.com.ua/eng/news/2023/01/6/7383768/ (accessed on 3 January 2023).

26　The text of the Tomos is available in English on the official website of the Patriarchate of Constantinople: https://ec-patr.org/patriarchal-and-synodal-tomos-for-the-bestowal-of-the-ecclesiastical-status-of-autocephaly-to-the-orthodox-church-in-ukraine/ (accessed on 3 January 2023).

27　https://www.asianews.it/news-en/The-Russian-Orthodox-against-the-claims-of-Constantinople-54091.html (accessed on 3 January 2023).

28　Moscow, on the other hand, which originated around 1147 as a military outpost of one of the principalities into which Rus' was divided, became influential many years later; thus, Kyiv did not come under Moscow's control until 1667.

29　https://www.kmu.gov.ua/bills/proekt-zakonu-pro-zaboronu-moskovskogo-patriarkhatu-na-teritorii-ukraini (accessed on 3 January 2023).

30　https://zakon.rada.gov.ua/laws/show/2662-19#Text (accessed on 3 January 2023).

31　On 17 June 2022, the European Commission published its opinion in favour of granting Ukraine official candidate status; on 23 June 2022, in Brussels, the European Council granted Ukraine candidate status.

32　"Article 10—Freedom of thought, conscience and religion. 1. Everyone has the right to freedom of thought, conscience and religion. This right includes freedom to change religion or belief and freedom, either alone or in community with others and in public or in private, to manifest religion or belief, in worship, teaching, practice and observance".

33　The EU Charter of Fundamental Rights entered into force with the Treaty of Lisbon on 1 December 2009. It is legally binding in all EU member states.

34　On 7 February 2019, the Verkhovna Rada (the Ukrainian Parliament) amended the Constitution by a majority vote, supplementing it with the following formula that makes explicit the values behind its adoption: "[ . . . ] caring for the strengthening of civil harmony on Ukrainian soil, and confirming the European identity of the Ukrainian people and the irreversibility of the European and Euro-Atlantic course of Ukraine, striving to develop and strengthen a democratic, social, law-based state [ . . . ]". https://www.refworld.org/pdfid/44a280124.pdf (accessed on 3 January 2023).

35　https://www.rainews.it/ (accessed on 4 January 2023).

36　Convention (IV) respecting the Laws and Customs of War on Land and its annex: Regulations concerning the Laws and Customs of War on Land. The Hague, 18 October 1907: "Art. 27. In sieges and bombardments all necessary steps must be taken to spare, as far as possible, buildings dedicated to religion, art, science, or charitable purposes, historic monuments, hospitals, and places where the sick and wounded are collected, provided they are not being used at the time for military purposes. It is the duty of the besieged to indicate the presence of such buildings or places by distinctive and visible signs, which shall be notified to the enemy beforehand".

37    It should be noted that the 1899 and 1907 Conventions do not differ with regard to the protection of cultural property in the event of armed conflict.

38    https://www.euronews.com/culture/2022/03/18/italy-ready-to-rebuild-bombed-mariupol-theatre (accessed on 21 December 2022).

39    https://culturalrescue.si.edu/ (accessed on 21 Dcember 2022).

40    UNESCO *Convention for the Safeguarding of the Intangible Cultural Heritage*, adopted by the UNESCO General *Conference* on 17 October 2003.

41    UNESCO *Convention for the Protection and Promotion of the Diversity of Cultural Expressions*, adopted in Paris on 20 October 2005.

42    "Article 5. A Member of the United Nations against whom preventive or enforcement action has been taken by the Security Council may be suspended from the exercise of the rights and privileges of membership by the General Assembly upon the recommendation of the Security Council. The exercise of these rights and privileges may be restored by the Security Council»; «Article 6. A Member of the United Nations who has persistently violated the Principles contained in the present Charter may be expelled from the Organisation by the General Assembly upon the recommendation of the Security Council".

43    https://www.unesco.org/en/articles/mapping-host-countries-education-responses-influx-ukrainian-students (accessed on 17 December 2022).

44    Convention for the Safeguarding of Intangible Cultural Heritage, Art. 16—*Representative List of the Intangible Cultural Heritage of Humanity*.

45    *Ibid*, reference is made in particular to the following words: "the Committee, upon the proposal of the States Parties concerned, shall establish, keep up to date and publish a Representative List of the Intangible Cultural Heritage of Humanity". This quote was contained in Art. 16.1, which excluded an attribution of national authorship to the candidate object.

46    https://www.unesco.org/en/articles/unesco-president-zelensky-officially-announces-odesas-candidacy-receive-world-heritage-status (accessed on 21 December 2022).

47    The new 1954 Hague Convention considered two distinct cases, general protection and special protection, which did not differ in the protection mechanisms adopted in concrete application. Paradoxically, the granting of the special protection regime provided for a longer and more cumbersome procedure that slowed down recognition and exposed the at-risk assets to greater dangers.

48    We refer, in this context, to the case of water extinguishing systems in fire situations in US museums, which are not provided in European ones. This needs to be specified when drawing up the Standard Facilities Report on the occasion of the lending of works (Manoli 2015).

49    Second Protocol to Hague Convention of 1954 for the Protection of Cultural Property in the Event of Armed Conflict, The Hague, 26 March 1999, arts. 10–13; art. 24.

50    Here we speak of individual criminal liability, which is common to all the articles of Chapter 4, including the first one, Article 15, which begins "Any person [ . . . ]".

51    "Article 38—State responsibility. No provision in this Protocol relating to individual criminal responsibility shall affect the responsibility of States under international law, including the duty to provide reparation".

52    UN Doc. S/RES/2253 (2015), 17 December 2015.

53    *Ibid*, "Condemning the destruction of cultural heritage in Iraq and Syria particularly by ISIL and ANF, including targeted destruction of religious sites and objects; and recalling its decision that all Member States shall take appropriate steps to prevent S/RES/2253 (2015) 15-22456 5/28 the trade in Iraqi and Syrian cultural property and other items of archaeological, historical, cultural, rare scientific, and religious importance illegally removed from Iraq since 6 August 1990 and from Syria since 15 March 2011, including by prohibiting cross-border trade in such items, thereby allowing for their eventual safe return to the Iraqi and Syrian people".

54    Charter of the United Nations. Chapter VII—Action with Respect to Threats to the Peace, Breaches of the Peace, and Acts of Aggression: "Article 39. The Security Council shall determine the existence of any threat to the peace, breach of the peace, or act of aggression and shall make recommendations, or decide what measures shall be taken in accordance with Articles 41 and 42, to maintain or restore international peace and security".

55    More information on the initiatives promoted by the organisation can be found at https://linktr.ee/ukrainedao (accessed on 23 December 2022).

56    https://donate.thedigital.gov.ua/nft (accessed on 22 December 2022).

57    https://metahistory.gallery/ (accessed on 18 December 2022).

58    It is recalled that the conversion of a work of art into an NFT consists of applying an authentic digital signature to the work and giving it unique data (name of the owner, purchaser, etc.) compared to other existing copies. When one buys an NFT representing a work of art, you do not receive a physical copy. Most of the time, anyone can download a copy of a digital file for free. The NFT only represents the certificate of ownership, which is registered in a blockchain so that no one can tamper with it. While the owner of the token owns the original digital artwork, the creator of the NFT retains copyright and reproduction rights. During the European Council on 23 June 2022, EU leaders granted candidate country status to Ukraine, which implies that the country will have to comply with relevant EU guidelines, including Art. 14 of Directive (EU) 2019/790 of the European

Parliament and of the Council of 17 April 2019 on copyright and related rights in the Digital Single Market, according to which "Member States shall provide that, when the term of protection of a work of visual art has expired, any material resulting from an act of reproduction of that work is not subject to copyright or related rights, unless the material resulting from that act of reproduction is original in the sense that it is the author's own intellectual creation". In other words, according to the European legislator, faithful reproductions of these works should not be protected. See also the Policy Department for Citizens' Rights and Constitutional Affairs Directorate-General for Internal Policies (2022).

59   https://metahistory.gallery/about-us (accessed on 21 December 2022). The inclusion of cryptocurrency among the weapons supporting Ukraine was made possible by the explosion of the NFT phenomenon in 2021, which invaded the art market following the Christie's auction, where the token of Mike Winkelmann's work, *aka* Beeple, *Everyday—The First 5000 days,* sold for USD 69.3 million. Riding on the trend of cryptocurrencies, the first museum was opened in Seattle in which exclusively NFT works are exhibited. The same technique was used to guarantee a preventive form of protection for Web artists, but also to create art to be exhibited to the public sic et simpliciter, indirectly proposing a new interpretation of Duchamp's ready-made philosophy: an object takes on the value of a work of art when the artist chooses to exhibit it as such in an environment designed for public enjoyment. At the same time, this museographic choice implies a criterion of exhibition that does not concern the contemplation of the work of art for its contents of interpretation and social criticism, but rather its forms of fruition and circulation in the market world. With the contribution of many artists who have chosen to turn to non-traditional media to convey their forms of expression since the second half of the 20th century, contemporary currents have developed in which the choice of medium becomes an integral part of the creative process of the work and completes its interpretation. Suffice it to think of an artist of international renown, such as Bill Viola, who exploits multimedia to produce something that was not there before, making it impossible to separate the work from its medium at a time when the medium itself, namely, video art, constitutes the discriminating factor in identifying the artistic strand in question. Following the same theoretical criterion, NFT technology also becomes a production apparatus, carving out an exclusive collector's market area and its own field of research within the study of art criticism (Paone 2018, p. 26).

60   On August 24th, in the framework of the 26th ICOM General Conference held in Prague, the ICOM Extraordinary General Assembly approved a new museum definition, stating: "A museum [ . . . ] exhibits tangible and intangible heritage". For the official and full definition, see https://icom.museum/en/news/icom-approves-a-new-museum-definition/ (accessed on 3 January 2023).

61   This is the case of the affair of Michelangelo's Tondo Doni kept in the Uffizi Gallery in Florence, the digital version of which was sold to a private collector for a sum of EUR 240,000 and helped open the doors for NFTs into the world's most famous museums, including the Hermitage in St. Petersburg, which inaugurated a project similar to the Italian one with the aim of "making luxury more accessible" (words spoken by the Hermitage museum's director, General Piotrovsky (Roccella 2021). This has triggered an alarmed reaction from the General Directorate for Museums of the Italian Ministry of Culture, which has requested the suspension of contracts for the digitisation of museum collections pending the drafting of official guidelines in compliance with the law on copyright and reproduction rights, under which digital reproduction practices using blockchain technology should fall.

62   Coined expression for the sale of museum collections to obtain immediate liquidity (Jandl and Gold 2021).

63   A digital cultural asset is defined as "a documentary object bearing images and knowledge with cultural content drawn from the intangible dimension of the basic one". The realisation of a digital counterpart of an already existing good does not imply for the legislator the creation of a new cultural good or a new testimony with value to civilisation; digitisation is not a simple conversion of goods from analogue to digital format, but is an interpretation, alternative with respect to form and content, which implies a subjective and contextualisable contribution in space and time. (Forte 2019, pp. 245, 266).

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
