# Peer review of "Cultural Heritage and Religious Phenomenon between Urbicide and Cancel Culture: The Other Side of the Russian–Ukrainian Conflict"

_religions, doi:10.3390/rel14040535_

Round 1

Reviewer 1 Report

This is a well-referenced and interesting paper, yet currently should not be published for two interconnected reasons:

1. The paper contains two discussions which are connected but should be conducted in two different papers: The religious conflict within Ukraine should be separated from the issue of Russia's military attack on Ukraine. While there is a loose connection between the two, it is confusing to see them debated within one paper.

2. The complexities of Russian Orthodoxy in general and in Ukraine in particular are incompletely discussed. Some basic facts about international Orthodox Christianity and the emergence of the Russian Orthodox Church and its arm in Ukraine are missing. Above all, the Orthodox churches are organized nationally accross the world. The UOC-MP was thus always a strange exception. Moreover, this misnamed Church which was actually not a proper Ukrainian church is the arm of the ROC which is specific by its close connection to the Russian state since the time of Peter the Great. Moreover, the current Moscow Patriarchy was created in 1943 by Stalin. That the ROC then supported the Russian war against Ukraine since 2014 is not that much of a suprise and neither is the enhanced drive for autocephaly in Ukraine since then.

The author should perhaps carve out the part about cultural heritage from the paper and submit a new text. He/she may then do some additional reading on Orthodoxy, and could then resubmit a significantly updated version of this second part as a separate paper.

Author Response

I have tried to clarify the goal of the article and to better explain some unclear concepts as from indications.

Reviewer 2 Report

It is worth continuing the research by taking into account the international context, such as the actions of other countries to save Ukraine's cultural assets. Many works of art are kept by Ukrainians in Poland. For example, paintings from Lviv have found their way to the National Museum in Poznan. https://www.poznan.pl/mim/wortals/en/en/news,9560/caution-risk-of-destruction,194030.html

The category of 'sacral geopolitics', which is pursued by Russia in the name of the idea of 'Moscow - the Third Rome' and 'Holy Rus', is also worth considering in future articles.

https://doi.org/10.1093/acrefore/9780190201098.013.1243

https://www.jstor.org/stable/41050783

https://doi.org/10.12775/HiP.2018.010

Author Response

I have tried to clarify the goal of the article and to better explain some unclear concepts.

I am deepening the research for another article and I am taking into account your indications. Thank you very much!

Reviewer 3 Report

This paper is problematic.

The argument is convoluted.

On the one hand it raises an important question about inadequacy of the international law to protect the cultural heritage in the interstate conflict. This conversation, while complicated, is warranted. 

Yet, the rest of this paper seems to be more about the concept of cultural heritage and art, where the author lamest the application of cancel culture to the Russian arts and culture in Ukraine and elsewhere in light of Russian war in Ukraine. 

The concept of urbicide is curios, yet underdeveloped.

The discussion of religion is confusing.  The author seems to argue that Orthodoxy is part of both countries broader cultural history. However, the treatment of the questions of religion  and religious freedom are skewed in this interpretation.  Church in this context is more than cultural heritage. MP is an active participant in the conflict, with plenty of evidence to that affect.  

Some of discussion of "evidence" seems to be informed by limited view of limited sources. For instance lines 279-281. What is the source for this information? This is a gross assumption. 

Author Response

(The authors gave the same response as above.)

Round 2

Reviewer 3 Report

The abstract was revised, however, the paper was not. 

Author Response

I have inserted the bibliography of points 279-281 (now 291-293). Throughout the text there are connections (reasoned and justified by the bibliography) between the role of the Orthodox (and the relative conflict within it which mirrors the Russian demands in the defense of the Russian religious tradition of the Moscow Patriarchate, and the autonomistic Ukrainian ones which claim their Church, no longer tied to Moscow and even accepted by the Patriarchate of Constantinople). From a religious point of view, this circumstance justifies two "warlike behaviors": the Russian urbicide of churches and places of worship of the Ukrainian Church, and the cancel culture of a religious nature on the part of Ukraine. The patriarchate of Moscow is certainly an active part of the conflict and precisely for this reason the religious cultural heritage is damaged. Religion is part of the conflict and destroying churches (and persecuting the Ukrainian clergy) is a clear demonstration of this. International law and the work of UNESCO appear to be inadequate to protect cultural and religious heritage. One solution could be that of "digital cultural heritage" if only to keep alive the memory of what existed before the conflict, including the religious memory testified by the destroyed buildings of worship. This is very important for building a new identity which maintains its roots and which is an effective expression of the will of the people and not just a political strategy.
I don't understand what is not clear there and where such links are lacking.
By modifying the abstract, I have once again given the interpretation of the paper. I ask please what is still not clear and where to intervene.